# A Possible Perspective of Recultivation with Arbuscular Mycorrhiza-Inoculated Drought-Tolerant Herbaceous Plants

**DOI:** 10.3390/plants12244088

**Published:** 2023-12-06

**Authors:** László Papp, Akale Assamere Habtemariam, Sára Brandt, Péter Cseh, Ádám Heller, Balázs Péter, Ágnes Pappné Szakály, Péter Kiszel, Borbála Codogno, Zoltán Bratek, Zoltán Tóth

**Affiliations:** 1Botanical Garden, Eötvös Loránd University, 1083 Budapest, Hungary; szakaly.agi@gmail.com (Á.P.S.); kiszel@fuveszkert.elte.hu (P.K.); codogno.b@gmail.com (B.C.); 2Department of Plant Physiology and Molecular Plant Biology, Eötvös Loránd University, 1117 Budapest, Hungary; akale@student.elte.hu (A.A.H.); brandts@freemail.hu (S.B.); pet321@student.elte.hu (P.C.); lpxjpe3@student.elte.hu (Á.H.); citywarrior@student.elte.hu (B.P.); bratek.zoltan@ttk.elte.hu (Z.B.); 3Department of Plant Systematics, Ecology and Theoretical Biology, Eötvös Loránd University, 1117 Budapest, Hungary; toth.zoltan@ttk.elte.hu

**Keywords:** regionality, Nagykunság, herbaceous, seed, mixture, germination, arbuscular mycorrhiza, *Rhizophagus irregularis*, inoculation

## Abstract

Using native species for urban green space is rather important nowadays. Plant cover on soil is necessary for agronomical and architectural investments as well as conservational programs, which all need minimal maintenance and have to be cost efficient. Commercially available seed mixtures for grasslands and lawns include species that partly originated from other mesoclimatic zones, and thus they may not be able to survive in the long-term, nor will they be adventive to the local ecosystem. With a focus on climate change, the most arid part of the Pannon geographical region was selected (near Törökszentmiklós in Nagykunság, Hungarian Great Plain). The local flora has adapted effectively to the environment; therefore, many species growing there were candidates for this study. Annuals and herbaceous perennials were investigated with respect to harvestability, reproducibility, decorativity, seed production, seed morphological characters (size, mass) and germination features. The selected 20 taxa were inoculated with INOQ Agri mycorrhiza (*Rhizophagus irregularis*) to increase the drought tolerance and biomass of the plants. Mycorrhizal frequency was significantly different among the taxa, reflecting various responses to the symbiotic interaction and possibly various mycorrhizal dependence of the plant species examined. We did not observe significantly higher colonization rate in most cases of the samples with artificial inoculation treatment. We conclude that the degraded mowed lawn soil that we used could contain propagules of AM fungi in a sufficient amount, so in the artificial grassland restorations, the additional AM inoculation treatment is not necessary to achieve a higher AM colonization rate.

## 1. Introduction

Recultivation of areas that previously lost their natural vegetation cover due to human activity is one of the most important environmental issues of recent time. The aim is primarily to restore native vegetation, because these species are most effectively adapted to local environmental conditions and take part in ecological processes [1]. Nonetheless restoration is hampered by a number of factors, such as the presence of toxic substances in the soil [2] or the presence of invasive plant species [3]. The reintroduced vegetation may be able to gradually improve degraded soils or remove toxic substances; this approach is known as phytoremediation [2,4]. Under the continental climate of Hungary, one of the most common vegetation types is dry grassland [5]. Several attempts have been made in these areas, mainly to revegetate abandoned cultivated areas and degraded grasslands, using propagules from their own region [6,7]. Natural succession is initiated in abandoned areas, but in many cases, species-poor communities are established, often dominated by invasive plants, and the native community is restored only after a long period of time [8]. Successful sites for the establishment and spread of invasive plants are untreated or over-treated roadsides, curbs and linear structures [9]. The creation and preservation of (preferably natural) grasslands in these places have great importance, mitigate the effects of stress factors as a buffer zone [10], and mostly in the case of native species, provide a propagule and food source, and can be a decorative sight for road users [11]. Therefore, experiments are underway to find ways to effectively introduce native species to maximize their chances of forming long-term vegetation and avoid their displacement by invasive species [7].

There are many methods of artificial grassland restoration. Spontaneous succession often will start naturally, but in many cases, it cannot happen if the place is too far separated from the plants’ natural habitats, or if non-indigenous invasive plant species start to take over the area. So, it is often needed to implement an artificial intervention to help to restore the natural conditions. These methods include sowing different seed mixtures, transfer of plant material, topsoil transfer, or translocation of the whole community. These methods have different success rates and costs, so it is necessary to choose the appropriate method [12,13].

Arbuscular mycorrhiza (AM), which is one of the oldest and most widespread types of mycorrhiza on Earth, may potentially have a remarkable positive impact on the process of recultivation. AM is a type of endomycorrhizal symbioses and belongs to the Glomeromycota division. In the case of arbuscular mycorrhizae, the hyphal cells penetrate the root cortex cells of plants and form nutrient-transferring formations, called arbuscules and hyphal loops [14]. Most terrestrial plant species are capable of forming arbuscular mycorrhizae, except for some plant families that are mostly non-mycorrhizal (e.g., Caryophyllaceae, Brassicaceae). They form ecto- or endomycorrhizae or form specific mycorrhizae with species of a particular plant family (e.g., Orchideaceae, Ericaceae) [15]. Most of the herbaceous plant species in Europe and Hungary have been shown to form arbuscular mycorrhizae [16,17], and in some cases, even in species whose family species have been studied so far and were not generally considered to form mycorrhizae [16]. Arbuscular mycorrhizae are generally considered to be a mutualistic symbiosis that benefits both organisms involved [15,18]. There are several published studies that discuss the physiological changes caused by the presence of AM in plants, as well as its correlations with various environmental factors. Previous studies have shown that the extent of AM colonization is significantly influenced by the available soil nutrient content [14], and also pointed out that under significant stress, the survival rate of colonized plants is higher, and certain physiological parameters are positively affected (e.g., increased growth) [19]; thus, AM plays an important role in the adaptation of plants to adverse conditions [18].

Upon recognition of these positive effects of AM, artificial mycorrhization of plants has become increasingly widespread in agriculture, horticulture and nature conservation. The use of AM fungi inoculums helps to successfully recover degraded, vegetation-deprived areas and to establish viable vegetation [19].

The relationship between AM colonization and environmental effects has been investigated in several cases in different disturbed habitats, such as heavy metal-contaminated soils [20], and in habitats with different successional stages [21]. In Hungary, research has shown that the rapidly spreading invasive plants are typically not mycorrhizal, or they have low mycorrhizal dependency [22]. Other plant species have been shown to have lower levels of mycorrhizal colonization in invasive populations in areas where the species is introduced than in native populations [23]. Additionally, experiments in abandoned mines have shown that higher mycorrhizal rates resulted in higher survival rates for most plant species [2]. Colonization by AM fungi also affects seed production, seed size and germination [24]. However, no targeted studies have yet been conducted on whether and to what extent artificial mycorrhization can assist in the re-establishment of dry grassland communities. The main purpose of the present study is to expand our knowledge of this field, which also promises to be useful from the point of view of practical application.

## 2. Results

### 2.1. Germination Test

Mycorrhizal inoculation was performed for 20 species (Figure 1a–c). The average seedlings per pot were counted in three control and three mycorrhizal-inoculated pots per species, as presented in Table 1. In the case of five species (*Festuca arundinacea, Hypericum perforatum, Podospermum canum, Salvia austriaca, Salvia nemorosa*) there was a significant difference, and two of them, namely *Hypericum perforatum* and *Salvia nemorosa*, showed higher numbers of individuals as a result of mycorrhiza treatment.

### 2.2. Plant Morphometric Analysis

The arbuscular mycorrhizal symbiosis promotes and results in increased growth [24]. For comparison purposes, we investigated the stems (Table 2) and the leaves (Table 3). Figure 2 gives flavor to the appearance of plants. We observed significant differences in morphological characters of six species (in bold): for the number of leaves per species of *Echium vulgare*, *Plantago media*, *Podospermum canum*, *Salvia pratensis*, the control values were higher for *Echium vulgare* and *Podospermum canum*; the length of the leaves had significant higher control values for *Echium vulgare* and *Salvia austriaca*. The width of the leaves (Table 3) and the length of the stem (Table 2) showed no significant difference.

### 2.3. Root Investigations

In several examined plant species, the colonization level of the control and AM-colonized plants was almost the same. In many cases, it was higher in the control samples, while in only a few cases, it was higher for certain parameters in the inoculated plants. The estimated values of colonization parameters (frequency of mycorrhization (F%) referring to the proportion of roots that do not contain any mycorrhizal structures, intensity of mycorrhization (M%), arbuscule content of the colonized area (a%), arbuscule content of the entire root (A%), the vesicle content of the colonized area (v%), and the vesicle content of the entire root (V%)) can be seen in Table A1. We have summarized the frequency of mycorrhization (F%) (Figure 3) and intensity of mycorrhization (M%) results (Figure 4).

The results of the examined samples are discussed below by plant species. In the case of *Echium vulgare*, *Filipendula vulgaris*, *Pseudolysimachion spicatum*, *Salvia pratensis*, *Silene viscosa*, *Trifolium pratense* and *Verbascum phoeniceum,* we failed to grow enough plants, so we could not evaluate the AM colonization parameters.

*Ajuga genevensis*: the frequency (F%) and intensity (M%) of mycorrhizal colonization is relatively high (well above 60%); the content of the arbuscule of the entire root (A%) and the content of the arbuscule of the mycorrhized area (a%) is above 35% in the case of control plants, while A% and a% are below 20% in the case of inoculated plants. Among all the parameters, higher values were obtained only for v% in the inoculated samples.

*Centaurea jacea*: based on the estimations, the proportion of fungal structures is relatively high for both treatment groups. However, that of the arbuscules and vesicles is relatively low (below 60%), although vesicle content (v%) is higher in inoculated plants.

*Festuca arundinacea*: although the frequency (F%) and the mycorrhizal intensity (M%) are considerably higher in the control plants, a% and v% are slightly higher in the inoculated group.

*Festuca pseudovina*: the mycorrhizal frequency (F%) and intensity (M%) of the inoculated plants are higher, but the arbuscule (A%, a%) and vesicle (V%, v%) contents are much lower than that of the control plants.

*Festuca rupicola*: the values of F% and M% are higher in the control, while all the other values are slightly higher in the inoculated samples.

*Hypericum perforatum*: we estimated conspicuously higher values for the inoculated samples than for the controls for all parameters, except for the vesicle content of the colonized areas (v%).

*Lotus corniculatus*: the values of the parameters of the control plants are higher than those of the inoculated ones, except for v%.

*Melica transsilvanica*: the values estimated for the control group are slightly higher in the case of F% and M%, and much higher for the other colonization parameters than in the inoculated group (Figure 5).

*Plantago media*: v% and V% are much higher, a% and A% are slightly higher for the inoculated samples, while F% and M% are higher for the control specimens.

*Podospermum canum*: the difference between most colonization parameters of the control and the inoculated group is small, but v% and V% are much higher in the control group.

*Salvia austriaca*: F% and M% are slightly higher; a% and A% are much higher in the control group, while v% and V% are considerably higher in the inoculated group.

*Salvia nemorosa*: the values of the inoculated samples are conspicuously higher than those of the control samples in terms of all investigated parameters (Figure 5).

As for the species *Echium vulgare*, *Filipendula vulgaris*, *Pseudolysimachion spicatum*, *Salvia pratensis*, *Silene viscosa*, *Trifolium pratense* and *Verbascum phoeniceum*, only one specimen within a treatment survived or none survived in one of the two treatments, so we could not interpret their data.

## 3. Discussion

Our knowledge of AM colonization of drought-tolerant herbaceous plant species still needs to be expanded; nevertheless, during our work, we gained some new information regarding this topic.

With the exception of *Hypericum perforatum* and *Salvia nemorosa*, the examined root samples of the plant species obtained higher mycorrhizal data in the control group. These findings are supported by the germination and morphological results. In contrast to the results of previous studies [25,26], we observed higher mycorrhizal colonization levels for both aforementioned species. The inoculation resulted in a significantly higher rate of germination for *Festuca arundinacea*, *Hypericum perforatum*, and *Salvia austriaca*, and a higher number of leaves for *Plantago media* and *Salvia pratensis*, but the other values did not differ significantly for these species.

The reason for this is that the soil used during the study, similar to the soils of the areas of interest for the practical application of roadside vegetation, was not cleaned of naturally present spores, including the spores of AM-forming fungi. One of the reasons for the obtained difference may be that the arbuscular mycorrhizal fungi naturally present in the soil could colonize the plant roots more efficiently than the strains in the added inoculum. These results are in concordance with the outcomes of a previous study by Shao et al. [27]. In the case of *Centaurea jacea* and *Festuca pseudovina*, the observed mycorrhization frequency (F%) was higher, compared with the previous studies [12,28]. On the other hand, in the case of *Lotus corniculatus*, the observed mycorrhization frequency (F%) was slightly lower compared with a previous study [29].

Our other possible explanation is that the AM fungi present in the soil and in the added preparation might compete with each other; therefore, the mycorrhizal colonization level was lower in the case of plants treated with the artificial inoculum, which is a phenomenon already observed in the case of mycorrhiza-forming fungi [30] and specifically with the AMF fungi *Rhizophagus irregularis* [31].

Our final conclusion is that arbuscular mycorrhizal fungi naturally present in the soil may provide a high level of colonization for plants set out in the field. On this basis, taking into account the negligible differences in the amount of mycorrhizal structures (except for two currently involved species *Hypericum perforatum* and *Salvia nemorosa*), the financial costs and time-consuming processes associated with the inoculation, it is not worth using artificial inoculums. However, it only stands for the soil of the currently sampled area; further investigations are needed to support this recommendation in general.

## 4. Materials and Methods

### 4.1. Experimental Setups and Germination Experiment

For the plant material, we chose one of the driest regions of Hungary, Nagykunság, where the loess and salty steppes are characteristically dry and treeless habitats. The soil on the edge of the roads is highly saline due to the salting of the roads in winter, but moving away from the road, the salt concentration decreases and loess grasslands are formed here. We investigated the grasslands region between 2017 and 2021, and we have chosen 20 non-endangered native species, which can serve as ornamental plants for humans and be frequented by insects [11]. For plant identification, Király et al. 2009 [32] were used. The seeds and fruits were collected and cleaned from May to November in 2020. Twenty seeds were seeded per pot in three repetitions (pots) per treatment. For a practical approach, the soil was not sterilized in order to see what kind of mycorrhizal relationship developed in such degraded soil as a result of artificial inoculation. For inoculation INOQ, Agri mycorrhiza (*Rhizophagus irregularis*) were scattered 1 cm deeper than the seeds, as for inoculation with 2 dL per m^2^, which was 0.016 dL per pot. The seeds were covered with a layer of soil equal to their thickness. Germination was started on February 9th and was monitored twice a week until May 15th. Sowing was carried out in a sunken, unheated greenhouse.

The soil, which was covered by degraded and species-poor mowed lawn, originated from Törökszentmiklós (WGS84 47.116737, 20.474221). The previous land use may have been similar, which was documented in different maps (e.g., Habsburg Empire—Cadastral maps (19th century). We took samples from two different depths (0–20 cm; 20–40 cm) to measure at the Agricultural Instrumentation Centre at the University of Debrecen (Table 4). We used these two layers mixed together for our research.

The measured values were compared per treatment with a two-sample *t*-test (*p* < 0.05).

### 4.2. Morphological Studies

The plants were measured in June 2021, before the investigation of mycorrhizal colonization which involves root cutting. The examined species were perennials, so mostly they had only developed leaves or a single stem in the first year (e. g. *Hypericum perforatum*). The leaf number and size of the former and the stem length were suitable for comparison. The stem was measured from the root neck to the end of the shoot for each emerged individual. In the case of grass species (Poaceae), the length of the largest leaf was measured as the part of the stem on each individual. For the species which do not form stems, three leaves were randomly selected, and the length and width were measured for each individual that emerged. The measured values were compared per treatment with a two-sample *t*-test (*p* < 0.05).

### 4.3. Investigation of Mycorrhizal Colonization

Sampling took place in July 2021. The roots of the plants were roughly cleaned from the soil, cut off and placed in 50% ethanol, then stored at 4 °C until the staining of fungal structures.

The staining was based on a technique by Vierheilig et al. [33,34], modified based on the preliminary results experienced with our own samples. At least thirty thin roots, cut to 1 cm, were stained per sample. The involved thin root segments did not include the root tips or thick branches of the root system, and all of them could potentially contain arbuscules. The root samples were kept in 1.5 mL Eppendorf tubes during each step of the staining, and the necessary compounds were added to them. First, we heated the samples in 10% KOH at 95 °C in a shaking block heater for 2 min while shaking. After that, we checked the softness of the roots by touching them with our fingertips. Those whose surface still appeared rough were heated for a further 1 min. Root samples removed from KOH were placed in a strainer and rinsed under running tap water (and the Eppendorf tubes were thoroughly rinsed). A 5% ink dye was used for staining: 87 mL of distilled water, 5 mL of Parker blue ink (ICO Ltd., Hong Kong) and 8 mL of 99–100% acetic acid for a final volume of 100 mL. After adding the dye, the samples were manually shaken vigorously and placed into the heater block for 5 min at 95 °C while shaking. To wash out the excess dye, the roots were rinsed again under tap water; then they were placed in 10% acetic acid and manually shaken. This was repeated three times, each time changing the 10% acetic acid in the Eppendorf tubes and rinsing the roots with water between each step; the second time the samples remained in acetic acid for 5, the third time for 10, and finally for 20 min at 30 °C while shaking. After another tap water rinse, the samples were placed in concentrated glycerol and stored at 4 °C until the microscopic slides were prepared. For the preparation, the 30 seemingly most intact roots per sample were placed on a glass slide and mounted in concentrated glycerol. The microscopic examination was carried out with Nikon Optiphot-2 and Olympus CH2 research microscopes. The level of colonization was estimated according to the methods of Trouvelot et al. (1986) [35]. For the evaluation, the following parameters were calculated: frequency of mycorrhization (F%) referring to the proportion of roots that do not contain any mycorrhizal structures, intensity of mycorrhization (M%), arbuscule content of the colonized area (a%), arbuscule content of the entire root (A%), the vesicle content of the colonized area (v%), and the vesicle content of the entire root (V%). The values of each species were averaged per treatment (control and inoculated).

## Figures and Tables

**Figure 1 plants-12-04088-f001:**
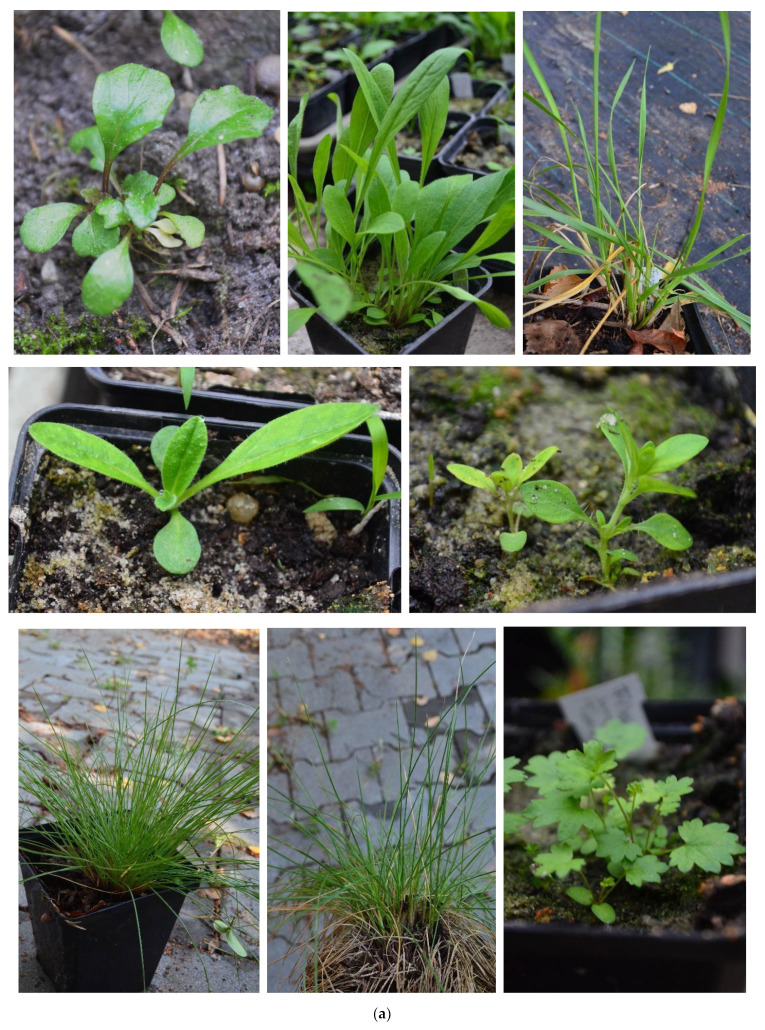
(**a**) The germinated seedlings are in pots in the greenhouse or planted in the ground. In the first row: *Ajuga genevensis*, *Centaurea jacea*, *Festuca arundinacea*; second row: *Echium vulgare*, *Hypericum perforatum*; third row: *Festuca pseudovina*, *Festuca rupicola*, *Filipendula vulgaris*. In these pictures, the plants are three months old, except for the *Festuca arundinacea* and *F. rupicola*, which are seven months old. (**b**) The germinated seedlings are in pots in the greenhouse or planted in the ground. In the first row: *Lotus corniculatus*, *Melica transsilvanica*, *Plantago media*; second row: *Podospermum canum*, *Pseudolysimachyon spicatum*, *Salvia austriaca*; third row: *Salvia nemorosa*, *Salvia pratensis*. In these pictures, the plants are three months old, except for the *Melica transsilvanica*, which are seven months old. (**c**) The germinated seedlings are in pots in greenhouse or planted in the ground. In the first row: *Silene viscosa* (two pictures), *Thymus glabrescens*; second row: *Trifolium pratense* and *Verbascum phoeniceum,* which was planted in the experimental plot. In these pictures, the plants are three months old, except for the *Silene viscosa* (1st picture), *Thymus glabrescens* and *Verbascum phoeniceum*, which are seven months old.

**Figure 2 plants-12-04088-f002:**
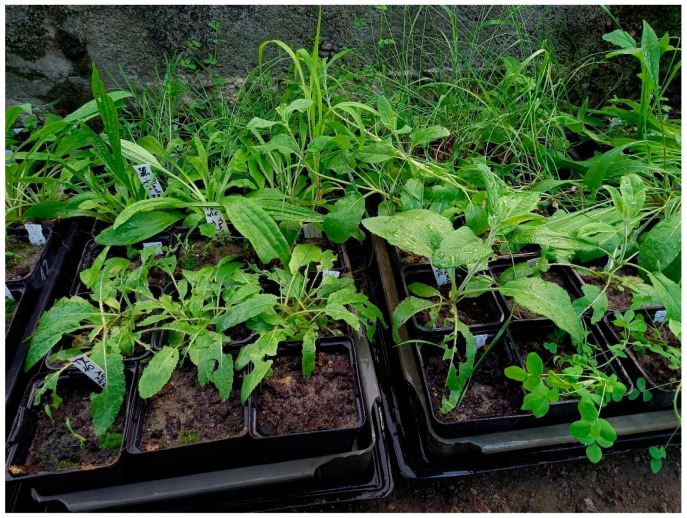
One part of prepared inoculated plants for measurement in two trays. The species are shown in three-pot groups per treatment. On the left tray, the following plants can be seen (from front to back): *Ajuga genevensis*, *Salvia pratensis*, *Thymus glabrescens*, *Plantago media*, *Lotus corniculatus*, *Podospermum canum*, *Festuca arundinacea*. On the right tray (from front to back): *Trifolium pratense*, *Salvia nemorosa*, *Centaurea jacea*, *Festuca arundinacea*. This figure is not suitable for presenting the results of the visible treatments, but is only an overview of the plant material at the time of measurement.

**Figure 3 plants-12-04088-f003:**
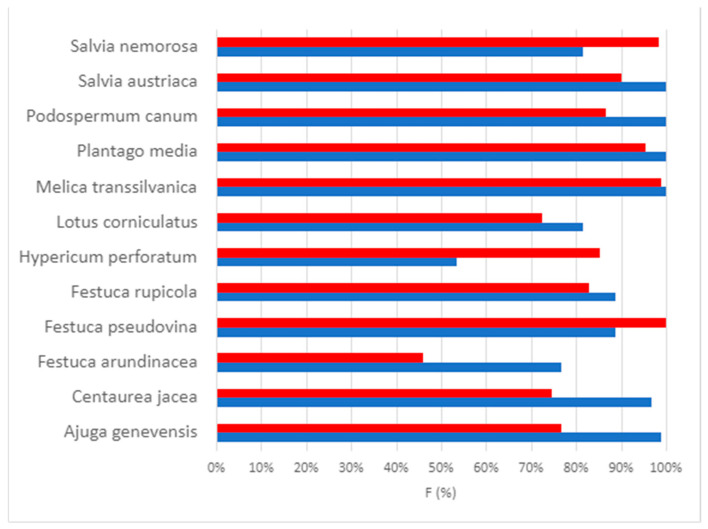
The estimated values of colonization parameters (frequency of mycorrhization (F%) referring to the proportion of roots that do not contain any mycorrhizal structures of the investigated control (blue) and inoculated plant species (red)).

**Figure 4 plants-12-04088-f004:**
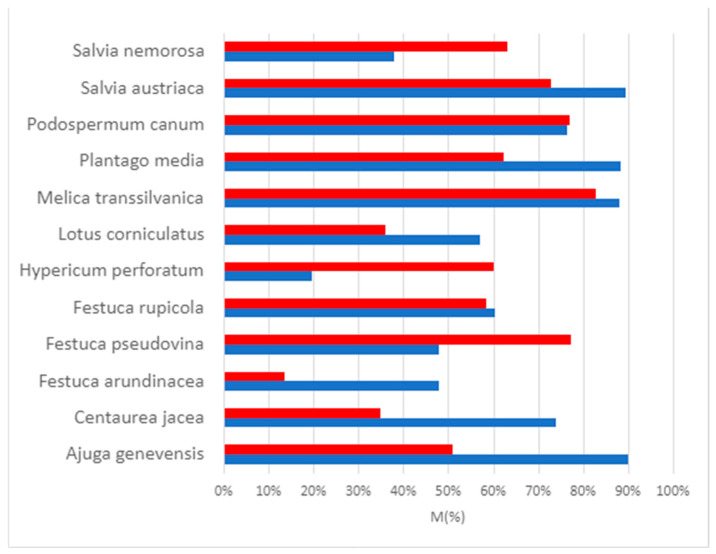
The intensity of mycorrhization (M%) of the investigated control (blue) and inoculated plants species (red).

**Figure 5 plants-12-04088-f005:**
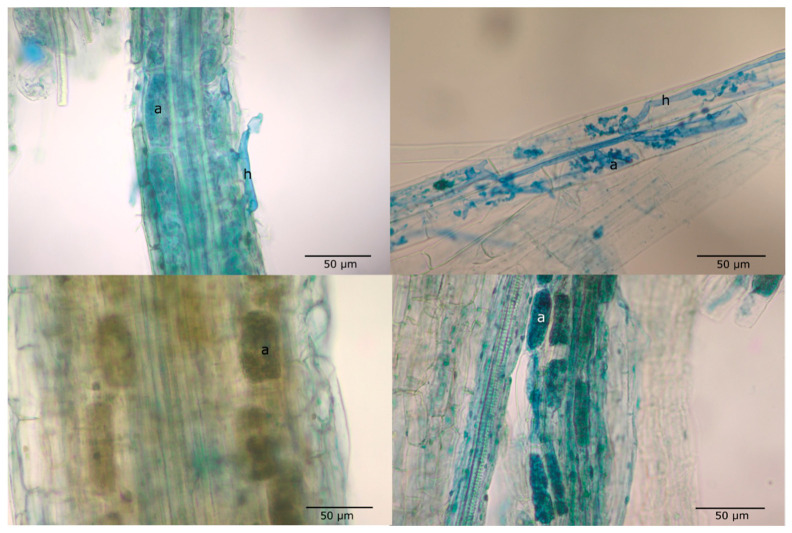
Microphotos on a root of a control (**upper left**) and an inoculated specimen (**upper right**) of *Melica transsilvanica*, and an inoculated (**bottom left**) and a control specimen (**bottom right**) of *Salvia nemorosa*, taken with Nikon Optiphot-2 research microscope, scale bars = 50 μm, arbuscules (a) and hyphae (h) are marked.

**Table 1 plants-12-04088-t001:** Results of the germination test. Treatments: control (C), AM inoculated (M). The origin column shows the collection point by settlement with the exact WGS84 GPS coordinate. The average germination per pot of 20 seeds (Aver. germ. per 20 seeds) with its standard deviation (SD germ. per 20 seed) and the germination percentage (Germ. percentage) were counted per treatment. The number of germinations was compared with a two-sample *t*-test. The bolded species names indicate a significant difference.

Taxon Name		Treatment	Origin	Aver. Germ. per 20 Seeds	SD Germ. per 20 Seeds	Germ. Perc. (%)	Pairwise *p*-Value < 0.05
1.	*Ajuga genevensis* L.	C	Abádszalók 47.401823, 20.677957	1.0	1.00	5	1
		M		1.0	1.73	5	
2.	*Centaurea jacea* L.	C	Abádszalók	12.0	3.00	60	0.208
		M	47.416913, 20.693190	9.0	1.73	45	
3.	*Echium vulgare* L.	C	Abádszalók	3.3	2.08	17	0.115
		M	47.412539, 20.689428	6.3	1.53	32	
4.	***Festuca arundinacea*** Schreb.	C	Törökszentmiklós	8.7	1.15	43	0.039
		M	47.123444, 20.480192	6.0	1.00	30	
5.	*Festuca pseudovina* Hack.	C	Törökszentmiklós	18.0	2.65	90	0.192
		M	47.124260, 20.482027	15.0	2.00	75	
6.	*Festuca rupicola* Heuff.	C	Törökszentmiklós	15.3	3.51	77	0.180
		M	47.124995, 20.483616	11.0	3.00	55	
7.	*Filipendula vulgaris* Moench	C	Abádszalók	8.0	3.00	40	1
		M	47.404759, 20.679292	8.0	5.00	40	
8.	***Hypericum perforatum*** L.	C	Abádszalók	1.0	1.00	5	0.016
		M	47.417316, 20.692507	3.7	0.58	18	
9.	*Lotus corniculatus L.*	C	Kunmadaras	4.0	2.00	20	0.609
		M	47.430432, 20.860312	3.3	0.58	17	
10.	*Melica transsilvanica* Schur	C	Törökszentmiklós	11.7	4.04	58	1
		M	47.124800, 20.482981	11.7	5.51	58	
11.	*Plantago media* L.	C	Törökszentmiklós	6.3	1.15	32	0.055
		M	47.122457, 20.478377	2.3	2.31	18	
12.	***Podospermum canum*** (C.A. Mey.) Griseb.	C	Kunmadaras	10.0	1.00	50	0.034
		M	47.430441, 20.860406	6.7	1.53	33	
13.	*Pseudolysimachion spicatum* (L.) Opiz	C	Abádszalók	1.0	0.00	5	0.374
		M	47.417119, 20.692839	1.3	0.58	7	
14.	***Salvia austriaca*** Jacq.	C	Tiszafüred	7.3	0.58	37	0.013
		M	47.591402, 20.851583	9.3	0.58	47	
15.	***Salvia nemorosa*** L.	C	Törökszentmiklós	9.7	2.52	48	0.011
		M	47.123603, 20.480440	2.7	1.15	13	
16.	*Salvia pratensis* L.	C	Tiszaszentimre	6.3	0.58	32	1
		M	47.468523, 20.804884	6.3	1.53	32	
17.	*Silene viscosa* (L.) Pers	C	Abádszalók	9.0	5.29	45	0.272
		M	47.416194, 20.690254	4.3	3.51	22	
18.	*Thymus glabrescens* Willd.	C	Abádszalók	0.7	0.58	3	0.519
		M	47.416850, 20.690566	0.3	0.58	2	
19.	*Trifolium pratense* L.	C	Törökszentmiklós	1.0	1.00	5	0.561
		M	47.123371, 20.480228	1.7	1.53	8	
20.	*Verbascum phoeniceum* L.	C	Abádszalók	0.7	0.58	3	0.116
		M	47.416492, 20.690413	0.0	0.00	0	

**Table 2 plants-12-04088-t002:** Results of the morphological analysis on the stem. Treatments: control (C), AM inoculated (M). Average stem lengths with their standard deviation (SD) and pairwise *p*-value were counted for average stem length by species.

	Taxon Name	Treatment	No. of Specimen	Average Stem Length (cm)	SD Stem Length	*p*-Value < 0.05	
1.	*Festuca arundinacea*	C	26	61.0	13.4	0.535	
		M	18	51.5	11.7		
2.	*Festuca rupicola*	C	46	30.6	4.9	0.462	
		M	33	28.5	7.2		
3.	*Festuca pseudovina*	C	54	20.5	4.1	0.286	
		M	45	26.8	5.1		
4.	*Melica transsilvanica*	C	35	25.4	5.5	0.668	
		M	35	26.7	5.0		
5.	*Lotus corniculatus*	C	12	29.7	12.5	0.630	
		M	10	33.9	9.2		
6.	*Hypericum perforatum*	C	3	14.2	10.9	0.700	
		M	1	19.1	10.2		

**Table 3 plants-12-04088-t003:** Result of the morphological analysis by leaves. Treatments: control (C), inoculated with mycorrhiza (M). Number of specimen (No. of specimen), average leaf number (Average leaf No.), average leaf length (cm) and average leaf width (cm) were counted with their pairwise *p*-values (*p* < 0.05) and standard deviations (SD). The bolded species names indicate a significant difference.

	Taxon Name	Treatment	No. of Specimen	Average Leaf No.	*p*-Value < 0.05	SD Leaf No	Average Leaf Length (cm)	*p*-Value < 0.05	SD Leaf Length	Average Leaf Width (cm)	*p*-Value < 0.05	Std. Deviation of Leaf Width
1.	*Ajuga genevensis*	C	3	6	0.116	0.6	4.3	0.862	0.7	1.5	0.775	1.3
		M	3	7		0	4.6		0.8	1.5		0.6
2.	*Centaurea jacea*	C	36	7	0.236	1.7	15.7		6.1	1.8		0.7
		M	27	7		1.7	15.5	0.823	5.6	1.9	0.811	0.6
3.	* **Echium vulgare** *	C	10	8	**0.000**	5.5	8.5	**0.011**	4	1.3	0.647	0.7
		M	19	4		1.3	3.7		2.6	0.8		0.3
4.	*Filipendula vulgaris*	C	25	9	0.719	6.2	3.7	0.142	2.1	1.4	1	0.6
		M	26	7		2.2	4.9		2.3	1.7		0.5
5.	* **Plantago media** *	C	18	6	**0.000**	2.3	8.1	0.021	4.8	1.6	0.385	0.6
		M	10	11		4.9	13.7		5.6	2.2		2.2
6.	* **Podospermum canum** *	C	30	9	**0.000**	4.3	16.2	0.032	4.9	0.4	0.164	0.2
		M	20	5		1.9	10.8		3.2	0.6		0.6
7.	*Pseudolysimachion spicatum*	C	3	16	0.607	6.1	5.8	0.189	1.7	2.4	0.487	0.7
		M	4	14		3.7	6		0.7	2.3		0.6
8.	* **Salvia austriaca** *	C	22	7	0.820	2.2	7.2	**0.007**	3.3	1.9	0.817	0.8
		M	28	7		2	4.9		1.9	1.4		0.7
9.	*Salvia nemorosa*	C	29	19	0.061	8.7	5.5	0.431	2.6	1.9	0.390	0.9
		M	8	14		7	5.7		3.2	1.6		0.9
10.	* **Salvia pratensis** *	C	17	7	**0.001**	14.7	5.6	0.515	2.6	1.8	0.797	0.9
		M	19	9		2	9		4.6	2.3		1.2
11.	*Silene viscosa*	C	27	10	0.206	3.7	6.3	0.422	3	1.8	0.580	0.8
		M	13	11		4.3	6.3		2.8	1.1		0.9

**Table 4 plants-12-04088-t004:** The results of the analytical soil test for the collected soil in different depths.

Elements	Soil Depth 0–20 cm	Soil Depth 20–40 cm
pH (KCl)	6.22 pH unit	6.42 pH unit
humus content	4.15 m/m%	2.51 m/m%
all water-soluble salts	0.05 m/m%	0.05 m/m%
Na	38.7 mg/kg	25.2 mg/kg
Mg	88.3 mg/kg	69.1 mg/kg
S	5.51 mg/kg	4.95 mg/kg
Zn	2.48 mg/kg	1.08 mg/kg
Mn	136 mg/kg	81.3 mg/kg

## Data Availability

The data presented in this study are available upon request from the corresponding author.

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
