# Peer review of "A Possible Perspective of Recultivation with Arbuscular Mycorrhiza-Inoculated Drought-Tolerant Herbaceous Plants"

_plants, 2023, doi:10.3390/plants12244088_

Round 1

Reviewer 1 Report (Previous Reviewer 2)

Comments and Suggestions for Authors

The authors have successfully responded to all the questions and suggestions that were raised in the initial review. Additionally, kindly take into account the following points for consideration:

  Minor issues:

Line 63: ´method.[33,34]´ the references are jumped not in ascending order. please check and rearrange. Eg. 33, 34 should be 12 and 13. 

Line 284, 285: 9th and 15th ´th´ should be superscript. 

Author Response

Thank you very much for your excellent comments. I modified it based on your suggestions.

Reviewer 2 Report (Previous Reviewer 3)

Comments and Suggestions for Authors

This paper had revised according to my suggestion, please accept in present form.

Author Response

Thank you again for your excellent work.

Reviewer 3 Report (New Reviewer)

Comments and Suggestions for Authors

The manuscript submitted for evaluation entitled  “A possible perspective of recultivation with AM inoculated drought-tolerant herbaceous plants” could be an interesting study if it were more refined. The topic addressed by the authors is important and necessary from the point of view of practical application, but the level of work is poor.

Comments:

- chaotic and too general description of tables with errors

 - incorrect order of cited authors in the paper (should be quoted from 1 to 34). After position 23 there is 30 instead of 24, what’s more some numbers are missing (16, 24, 27).

- the Discussion chapter is superficially discussed with other studies on this topic.

In my opinion, the manuscript can be accepted for publication if it is thoroughly improved.

Author Response

Thank you very much for your honorable work and thank you for your appreciative opinion on the manuscript. We will working on, that the level of work will be richer in the future. We corrected the errors in the tables. The incorrect order of cited authors and the absent numbers are corrected in the paper. We discussed the Discussion chapter with other studies on this topic.

Round 2

Reviewer 3 Report (New Reviewer)

Comments and Suggestions for Authors

The authors did not take into account all comments (e.g. inconsistency in the numbering of tables 2 and 3 in relation to the description in the text). The manuscript requires further correction (yellow marked in the text).

Author Response

Dear Reviewer,

thank you very much for your excellent comments. 

We changed the commas to dots in the "Table 1" and "Table 4".

We synchronized the serial numbers and the reference of the Tables in the manuscript.

We changed the "stems" and the "leaves" words in the text, the row of 146, which already corresponds to the serial number of the Table 2 and 3. 

We changed the reference the "Figure 3." to "Figure 5." in 219 and 228 rows.

We modified the Results and Discussion chapters for easier understanding.

I corrected and completed the "Author Contributions"chapter, because some names wroted not correct.

And we made minor spelling and stylistic changes in the text.

This manuscript is a resubmission of an earlier submission. The following is a list of the peer review reports and author responses from that submission.

Round 1

Reviewer 1 Report

Comments and Suggestions for Authors

Review remarks on the research manuscript entitled “Drought tolerant, diverse artificial grassland starting from mycorrhiza inoculated native seed mixture in a semi-arid region of Hungary” submitted by Papp et al. to plants-MDPI journal. The present article is not suitable for the publication in plants journal due to the lack of novelty of the research content.

Author Response

Dear Reviewer,

We would like to get  help from you to improve our manusctript. Many thanks!

Your sincerelly,

László Papp

Reviewer 2 Report

Comments and Suggestions for Authors

The article discusses the effects of mycorrhizal inoculation on various plant species in a semi-arid region of Hungary. The investigation seems to the interesting. However, there are a few areas where clarity and organization could be improved.

Major concerns:

Line 91: selected 20 species, but germination was shown using 2 species alone. It’s important for the readers to physically identify the species used in this study. Please include the photos for all the species in one single panel, Figure 1.

Table1: The column ´No. of germination percentage (%)´ is this data is No. Or %? You can not use both. For my understanding, it's % germination. If it is otherwise, your statistics will also change accordingly. Please clarify or correct. 

Figure 3: The figure legend is incomplete, and it's crucial for all figure captions to be self-explanatory and directly relevant to the experiment described. These changes apply to all the figures and tables of this submission. 

Line 103: 2.2 Plant morphometrical analysis. The results are not thoroughly elucidated within this segment. Nevertheless, there exists an opportunity to enhance this section. It's advisable to contemplate a revision.

Line 138: arbuscule of the entire root (A%)…. Based on what I've observed and my personal experience, it's quite uncommon for the entire root to be colonized. Usually, the root tips and young roots remain uncolonized. In this context, I recommend calculating the percentage of root colonization as outlined by McGonigle et al. in their publication "New Phytologist" (1990) or employing other established methodologies. However, if you happen to possess images of fully colonized root systems, I encourage you to include them, particularly if your observations reveal the presence of arbuscules throughout the entire root structure.

Lines 145, 172, 176, 178: ´The control group consists of only one sample, so it cannot be evaluated.´.. I believe that including such statements in the results might not be appropriate. If the sample volume is insufficient and hasn't been assessed, it's advisable to omit the inclusion of this species. Alternatively, if feasible, consider conducting the experiment again.     

Line 192-214: Consider discussing all your results along with the previous work. The authors used only one reference for the entire discussion. The discussion needs to be rewritten.

Line 244: Table 4 The results of the anatical soil test. Is not included in the results? Why? Nor discussed in the discussion?

Other important comments:

Lines 22 and 28: Rhizoglomus irregulare should be italicized.

Line 24: change ´The Inoculation was…… ´ The inoculation was ….´

Line 71-74: This paragraph should be merged with the above paragraph. 

Line 78-79: Instead of mycorrhizal, or mycorrhization I suggest using ´mycorrhizal colonized´

Line 84: Clarify what is Mycorrhizality?

Line 98: Table 1. None of the scientific names were italicized. 

In the origin column of Table 1 include the latitude and longitude for each collection site. Location name alone is not sufficient, and it is arbitrary.

Line 98: Result of germination test. Treatments: control (C), inoculated with mycorrhiza (M). What is the significance of (C) and (M) in parenthesis? clarify? It looks authors copied the legend from Table 2!. 

Figure 1-2: The legend is incomplete. You must mention how many days old these plants are. Conditions of the growth greenhouse or incubator? etc.

Line 115: Table 2. None of the scientific names were italicized. 

Table 2: ´The bolded species names indicate a significant difference.´…. I do not see any bolded letters!! Check and clarify.

Line 121: Table 3. Again! none of the scientific names were italicized.

Line 132: Values of the colonization… what do you mean by value? Root length colonization? quantitative data of the arbuscules? Or vesicles? Recommended to use technical words. 

Line 135: Table A3 should move to supplemental information. 

Line 137: Mycorrhizal intensity is high (M%)… please define high? Is it 90-100%? Make a scale (range) if you consider using high, medium, and low. 

Figure 4-7: Consider making a single panel using all the figures. The legends are incomplete. For all the figures scale bar needs to be included. All figures need to be labeled for the mycorrhizal structures like arbuscules, vesicles etc. Also, mention how many days post-inoculation these roots are. 

Figure 7: Control of Silene viscosa with arbusculums…. I do not see any arbuscules instead all are vesicles. 

Line 186: ´Figures´ change to ´Figure´. 

Comments on the Quality of English Language

English in the present manuscript is very difficult to understand. Especially in the abstract, results, legends of the figures and tables, methods and discussion. Nevertheless, the introduction has no such issues. 

Reviewer 3 Report

Comments and Suggestions for Authors

This paper title are not fit for the research owing to the AMF inoculation mix with orginal soil without sterilazation, so compare to control, AMF inoculation is much different in 20 taxa plants. In most experiment AMF seeds inoculation need soil sterilazation before sowing, the drought tolerance need soil moisture control. As the coloniation parameters are higher in CK(C) than imoculated with mycorrhiza(M), such results can not give reasonable explaination, so the results can not be accepted. 

Comments on the Quality of English Language English quality is good in this parper
